# Effects of β-1,6-Glucan Synthase Gene (*FfGS6*) Overexpression on Stress Response and Fruit Body Development in *Flammulina filiformis*

**DOI:** 10.3390/genes13101753

**Published:** 2022-09-28

**Authors:** Yuanyuan Liu, Xinbin Ma, Ying Long, Sen Yao, Chuanzheng Wei, Xing Han, Bingcheng Gan, Junjie Yan, Baogui Xie

**Affiliations:** 1Mycological Research Center, College of Life Sciences, Fujian Agriculture and Forestry University, Fuzhou 350002, China; 2Institute of Urban Agriculture, Chinese Academy of Agricultural Sciences, Chengdu 610213, China

**Keywords:** *F. filiformis*, β-1,6-glucan synthase, oxidative stress, injury stress, stipe elongation

## Abstract

β-1, 6-glucan synthase is a key enzyme of β-1, 6-glucan synthesis, which plays a vital role in the cell wall cross-linking of fungi. However, the role of the β-1, 6-glucan synthase gene in the development of the fruiting body and the stress response of macrofungi is largely unknown. In this study, four overexpression transformants of the β-1, 6-glucan synthase gene (*FfGS6*) were successfully obtained, and gene function was studied in *Flammulina filiformis*. The overexpression of *FfGS6* can increase the width of mycelium cells and improve the tolerance ability under mechanical injury and oxidative stress. Moreover, *FfGS6* gene expression fluctuated in up-regulation during the recovery process of mycelium injury but showed a negative correlation with H_2_O_2_ concentration. Fruiting body phenotype tests showed that mycelia’s recovery ability after scratching improved when the *FfGS6* gene was overexpressed. However, primordia formation and the stipe elongation ability were significantly inhibited. Our findings indicate that *FfGS6* is involved in regulating mycelial cell morphology, the mycelial stress response, and fruit body development in *F. filiformis*.

## 1. Introduction

*F. filiformis* (previously known as *F. velutipes*) is a typical agaric mushroom with a long stipe and a small pileus [1]. It is one of the most widely commercially cultivated edible mushrooms, with high dietary and medicinal value [2]. Similar to most edible mushrooms, the growth and development of *F. filiformis* are regulated by both genotype and environmental factors. Through differential gene expression, the external environment influences the development of fruiting bodies and the formation of agronomic traits [3]. Omics sequencing has built a substantial database of key regulatory genes involved in the stress response and fruiting body development of *F. filiformis* [4,5], yet the functional investigation of key genes is lacking. Genome and transcriptome analyses of *F. filiformis* by Liu et al. identified that *hsp70*, *hsp90*, and *fes1* may contribute to heat stress regulation in the early stage of fruiting body development [6]. Based on a comparative proteome analysis, Liu et al. revealed that 264 differentially expressed proteins in 176 pathways are responsible for the low-temperature and light stress responses of *F. filiformis* mycelium, suggesting a highly complex gene regulatory process [7]. Several studies have indicated that macrofungi lengthen their stipes by elongating their cell walls [8]. Additionally, the cell wall is also the first structure to sense external changes and respond to cellular stress [9,10,11,12].

β-1,6-glucan is an integral component of fungal cell walls [13]. It is synthesized from the endoplasmic reticulum, progresses through the Golgi apparatus, and reaches the cell surface [14,15]. Zhou et al. [16] reported that the heterologous expression of *GsmA*, a newly identified β-1,6-glucan synthase in *Mycoplasma agalactiae*, significantly increases the amount of β-glucan in yeast. In *Candida albicans*, the simultaneous knockout of two β-1,6-glucan synthase genes, *KRE6* and *SKN1,* results in abnormal cell walls, reduced mycelial growth, halted biofilm formation, and the loss of virulence [17]. Oyama et al. [18] analyzed the β-1,6-glucanase gene in *Neurospora crassa* using gene mutation technology. The resultant mutant induces no noticeable phenotypic changes but is significantly less resistant to Congo red, SDS, and CTAB. Studies on *Agaricus bisporus* have illustrated that the proportion of glucan side chains linked by β-1,6-glycosidic bonds is increased markedly during the period of rapid stipe elongation [19]. It has been shown that the transcriptional level of the β-1,6-glucan synthase gene remains elevated after harvesting the fruit body [20,21].

By analyzing the transcriptome data of *F. filiformis*, we identified and characterized a gene related to the development of the fruit body. A sequence analysis and bioinformatics identification revealed that the gene named *FfGS6* could encode β-1,6-glucan synthase [22]. In this study, *FfGS6* overexpression transformants were generated through *Agrobacterium*-mediated transformation. By combining a differential gene expression analysis with a transformants phenotypic analysis, we elucidated the effects of *FfGS6* overexpression on the stress response and fruit body development of *F. filiformis*.

## 2. Materials and Methods

### 2.1. Strains and Culture Conditions

The *F. filiformis* transformation recipient strain Fv01 (derived by pairing monokaryotic strains Fv01-10 and Fv01-N) was deposited in the Fujian Edible Fungi Engineering Technology Research Center and the National Edible Fungi Breeding Center. The binary expression backbone vector pBHg-eGFP (accession number: MZ420392) was provided by the Mycological Research Center of Fujian Agriculture and Forestry University. Competent *Escherichia coli* (DH5α), a TIANprep Midi Plasmid Kit II, and a Universal DNA Purification Kit were procured from Tiangen Biotech Co., Ltd. (Beijing, China). *Agrobacterium tumefaciens* GV3101 competent cells were purchased from Biomed Gene Technology Co., Ltd. (Beijing, China).

### 2.2. Construction of FfGS6 Overexpression Vector

Using the *FfGS6* gene sequence (accession number: MF457899) in the Fv01-10 genome as a template, Shanghai Jingran Biotechnology Co., Ltd. (Shanghai, China) synthesized target fragments with *Spe* I and *Apa* I restriction sites at both ends. The skeleton vector pBHg-eGFP was digested and linearized with *Spe* I and *Apa* I enzymes (TaKaRa Bio Inc., Otsu, Shiga, Japan); the target sequence and the linearized vector were connected with T4 DNA ligase (TaKaRa Bio Inc., Otsu, Shiga, Japan) to construct the *FfGS6* gene overexpression recombinant vector pBHg-*FfGS6*-OE (accession number: MZ374067). Once the recombinant vector was transformed into *E. coli*, the positive clones were verified by Sanger sequencing using the pFfGS6-F/pFfGS6-R primer mix. The strategy for vector construction is depicted in Figure 1.

### 2.3. Agrobacterium-Mediated Transformation of the FfGS6 Overexpression Vector into F. filiformis Mycelium

According to the methods illustrated by Okamoto et al. [23], *Agrobacterium*-mediated transformation was carried out with the Fv01 mycelial block of *F. filiformis* as the host. After the heat shock transformation of the constructed expression vector pBHg-*FfGS6*-OE into *Agrobacterium* GV3101, Sanger sequencing confirmed monoclonal colonies. Young *F. filiformis* mycelial blocks were excised from the CYM plate medium with a 6 mm sterile puncher, and they were co-cultured with *Agrobacterium*-harboring plasmids in IM induction medium for six hours [24]. Subsequently, the mycelial blocks were co-cultured in cellophane-covered IM plates for 3–5 days. After that, the mycelial blocks were rinsed with sterile water containing cephalosporin (200 μmol/L) and cultured in CYM agar plates containing hygromycin (30 µg/mL) for 15–25 days after surface dehydration with sterile filter paper. Putative transformants were obtained by primary screening and then transferred to PDA medium (200 g potato, 200 g glucose, 20 g agar, 1 L water, natural pH) containing hygromycin (60 µg/mL) to perform secondary screening at least three times in order to obtain putative transformants with stable inheritance.

### 2.4. Validation of Putative Transformants

The total DNA was extracted using a modified CTAB method [25]. The corresponding vector was adopted as the positive control, and the wild-type strain Fv01 was applied as the negative control. The primers pFfGS6-F and pFfGS6-R were utilized to amplify the *FfGS6* gene in the plasmid and its upstream promoter, and downstream terminator partial fragments. The hygromycin-resistant gene fragment was amplified by employing primers Hpt-F and Hpt-R. The primers used here are listed in Table 1. The band size of the amplified product was detected by agarose gel electrophoresis. According to Liu et al. [26], the transformants were re-sequenced. The insertion site and copy number of the exogenous fragments in the genome were subsequently detected.

To verify target gene expression, the putative transformants and the wild-type strains were seeded in 90 mm cellophane-covered PDA plates using a needle-punching approach. After seven days of culture at 25 °C in the dark, the fresh mycelium was quick-frozen in liquid nitrogen for gene expression detection.

### 2.5. RNA Extraction and RT-qPCR Determination of Relative Gene Expression

Total RNA was extracted using an OMEGA E.Z.N.A. Plant RNA Kit (Omega Bio-Tek, Norcross, GA, USA). According to the manufacturer’s instructions, the qualified RNA was reverse-transcribed into cDNA using a TransScript^®^ All-in-One First-Strand cDNA Synthesis SuperMix for qPCR Kit (TransGen Biotech, Beijing, China). Gene expression detection was performed with the utility of a TransStart^®^ Top Green qPCR SuperMix Kit (TransGen Biotech, Beijing, China) on a real-time fluorescence quantitative PCR instrument (Bio-Rad CFX96, Hercules, CA, USA), with an annealing temperature of 60 °C and a cycle number of 40. With the glyceraldehyde-3-phosphate dehydrogenase gene (*FfGAPDH*) as an internal reference, quantitative PCR primers were designed based on Primer-BLAST online software (https://www.ncbi.nlm.nih.gov/tools/primer-blast/index.cgi?LINK_LOC=BlastHome (accessed on 5 June 2017) and synthesized by Tsingke Biotechnology Co., Ltd. (Tianjin, China). The primer sequences are displayed in Table 1. The relative expressions of the target genes were calculated using the 2^−ΔΔCt^ method [27].

### 2.6. Mycelial Growth Rate Assay

The activated mycelium was seeded in a PDA plate and cultured at 25 °C for three days, after which a ‘cross’ line was drawn on the bottom of the plate with the mycelial block as the center. Meanwhile, a start line was established at the junction of the colony edge and the cross to mark the growth starting point. After three days of additional culture, terminated lines were drawn at the edges of the colonies and photographed. The distance from the start line to the terminating line was measured, the daily growth rate of the mycelium was calculated, and nine biological replicates were assayed for each strain.

### 2.7. Microscopic Observation of Mycelial Cells

The activated mycelium was seeded on PDA plates inserted with sterile coverslips using a needle-punching approach. After climbing to half the coverslips, the mycelium was dyed with a fluorescent whitening agent (Sigma, IN, USA) and then visualized and photographed under a fluorescent microscope BX51 (Olympus, Tokyo, Japan). The width of septate mycelial cells with the clamp connection was counted using ImageJ software (National Institutes of Health, Bethesda, MD, USA). Data were measured for at least 120 cells per strain at the clamp connection site.

### 2.8. Scratching and Assessment of Mycelial Recovery

Scratching and collecting mycelial samples: Fv01 mycelium was seeded into cellophane-covered PDA plates and cultured at 25 °C in the dark. A ‘#’-like scratch was made on the mycelium using a blade when 2/3 of the plate was covered with the mycelium. The injured mycelium was cultured at 25 °C in the dark for 2, 4, 6, 12, and 24 h. The mycelia were quickly harvested and quick-frozen in liquid nitrogen for later use, with three biological repetitions for each treatment.

Assessment of transformants’ mycelial recovery: The mycelium was seeded on a PDA plate. After the plate was entirely covered with the mycelium, the surface of the aerial mycelium on the plate was carefully scraped off. After punching the edge of the plate with a puncher (6 mm in diameter), the mycelium was transferred to agar medium (20 g agar, 1 L water) and continually cultured while observing the recovery of the mycelium. During the test, 28 colonies were treated for each strain, and the recovery status of the mycelium was recorded every 12 h until all colonies germinated. The germination and density of the mycelium of different strains were subsequently compared.

### 2.9. Hydrogen Peroxide Treatment and Mycelial Tolerance Test

Hydrogen peroxide stress induction and sampling: Fv01 mycelium was seeded onto PDA plates using needle punching, which contained 0, 1.25, 2.5, 5, and 10 mmol/L H_2_O_2_, and cultured at 25 °C in the dark. The diameters of the colonies on the sixth day of culture were measured and recorded by employing the crossline method, with nine biological repetitions per treatment. Meanwhile, the mycelium was harvested and quickly frozen in liquid nitrogen for RNA extraction and gene expression detection. Three biological repetitions were utilized for each sample.

Test of the transformants’ tolerance to H_2_O_2_: Fv01 was seeded on a PDA plate containing 5 mmol/L H_2_O_2_ using the needle-punching approach, and a PDA plate without H_2_O_2_ was used as a control. The crossline method was applied to evaluate the growth rate of the mycelium. The inhibition rate of H_2_O_2_ on mycelial growth was calculated as follows: [inhibition rate of mycelial growth = 1 − mycelial growth rate of the experimental group/mycelial growth rate of the control group]. Nine biological repetitions were performed for each strain.

### 2.10. F. filiformis Cultivation and Fruiting Body Phenotype Analysis

Referring to the method by Lee et al. [28], after the culture of the *F. filiformis* Fv01 mycelium mother strain and primary strain, the mycelium was seeded in a 350 mL glass bottle containing compost formulations (58% cottonseed husk, 20% sawdust, 20% bran, 1% glucose, 1% gypsum, and a water content of 63–65%) and cultured at 25 °C in the dark. After the bottle was filled with the mycelium, the mycelium was scratched and treated at a low temperature (8–10 °C) under weak light conditions to induce primordium formation. The bottle was bagged when the fruiting body grew out of the bottle, and the mycelium was cultured in the dark until it matured. Mycelium growth, mycelium recovery on the material surface after five days of scratching, and primordium formation were observed, photographed, and recorded. Additionally, the morphological differences of the fruit bodies were compared nine days after primordium formation. The stipe length of the 20 longest fruit bodies in each bottle was measured with five biological repetitions per strain.

### 2.11. Statistical Analysis

GraphPad Prism 8.0.2 software was adopted to plot bar charts. Data are presented in the form of mean and standard error. The differences between the treatment group and the control group were compared based on Student’s *t*-test, with * *p* < 0.05, ** *p* < 0.01, and *** *p* < 0.001 regarded as statistically significant. The Pearson correlation coefficient analysis was performed, employing SPSS v20.0 software.

## 3. Results

### 3.1. Acquisition and Validation of FfGS6 Overexpression Transformants

After primary screening using the hygromycin-resistant plate and multiple rounds of re-screening and verification, the sum of seven overexpression transformants with stable inheritance was generated; three monokaryotic strains were excluded through the observation of the clamp connection; the sub numbers of the remaining four overexpression transformants were OE#10, OE#12, OE#14, and OE#16. The specific primers Hpt-F/Hpt-R and pFfGS6-F/pFfGS6-R were employed to amplify the hygromycin-resistant gene sequences and the target gene sequences. Targeted bands with the same size as the positive controls could be produced after amplifying the four putative transformants (Figure 2a,b).

According to the detection results at the insertion site, all the exogenous fragments were single copies and stably inserted into these four strains, which were considered positive transformants with stable inheritance (Appendix A). The results of the RT-qPCR determination of the *FfGS6* gene expression in these positive transformants are summarized in Figure 2c. The expression levels of *FfGS6* in the overexpression transformants OE#10, OE#12, OE#14, and OE#16 were 10.07, 3.72, 3.86, and 4.39 times that of the wild-type strain Fv01, respectively, confirming the successful overexpression of the four *FfGS6* gene transformants.

### 3.2. Effects of FfGS6 Overexpression on the Growth of F. filiformis Mycelium

Subsequently, we detected the morphology and growth rate of the transformants’ mycelia (Appendix A). It was observed that the four overexpression transformants could grow normally on PDA plates and that their morphologies were not significantly different from that of the wild-type strain Fv01, all showing relatively dense aerial mycelia (Appendix A). The crossline method further measured the mycelial growth rate (Appendix A). The mycelial growth rates of Fv01 and the four transformants were 0.53, 0.64, 0.59, 0.63, and 0.48 cm/d. Among them, the growth rates of OE#12 and OE#16 were slightly slower, while those of OE#10 and OE#14 were slightly faster than that of Fv01. We speculated that the change in the mycelial growth rate might be an unexpected effect caused by the insertion of exogenous fragments and the overexpression of the *FfGS6* gene [26,29].

### 3.3. Impacts of FfGS6 Overexpression on the Microscopic Morphology of Mycelial Cells

Long et al. identified that *FfGS6* is a gene encoding the β-1,6-glucan of *F. filiformis*, which may be involved in the synthesis of cell walls [22]. To further clarify the role of *FfGS6* in cell morphogenesis, we adopted a fluorescent whitening agent to stain the mycelial cell wall, followed by microscopic observation. As depicted in Figure 3a, the four transformants and the wild-type strain Fv01 showed abundant clamp connection. The fluorescent whitening agent could effectively bind to the cell wall, through which the cells could be visualized in blue under ultraviolet excitation light. Accordingly, staining was observed in the septate mycelium with clamp connections to be significantly stronger than that in other areas. The cell width of the mycelial around the clamp connection was further measured, and the results are outlined in Figure 3b. The cell widths at the septate membrane of the transformants OE#10, OE#12, OE#14, and OE#16 were noticeably increased compared to that of Fv01. Combined with the potential function of *FfGS6* in the synthesis of cell walls, we speculate that the increase in cell wall thickness may be one of the reasons for the widening of the transformants cells.

### 3.4. The Role of FfGS6 in the Injury Stress Response of Mycelium

Dense scratching was utilized to induce the injury stress of Fv01 plate mycelium (Figure 4a), and the *FfGS6* gene expression was detected after 0, 2, 4, 6, 12, and 24 h of injury stress (Figure 4b). After 2 h of injury stress, the expression of the *FfGS6* gene was considerably up-regulated by 3.44 times, and it reached the peak at 4 h. Afterward, the *FfGS6* gene expression was reduced after 6 and 12 h and then increased again after 24 h. Overall, the results indicate an increasing, decreasing, and then increasing expression pattern. However, we found that the expression pattern of the *FfGS6* gene in normal strains did not change significantly in each time period (Appendix A). The results demonstrate that the *FfGS6* gene was engaged in the injury stress response. It is assumed that the *FfGS6* gene expression may be associated with cell wall regeneration and cell damage repair.

To further clarify the role of the *FfGS6* gene in mycelial injury repair, we evaluated the restoration ability of the four transformant mycelial blocks. As presented in Figure 4c, new mycelium was germinated in some blocks of all strains 12 h after injury induction, and the number of new mycelial blocks increased after 24 h. All mycelial blocks showed normal germination after 48 h. Compared to Fv01, among the four transformants, only OE#16 exhibited apparent advantages in the early stage of mycelial repair. In contrast, the germination rates of OE#10, OE#12, and OE#14 were markedly reduced. However, the germination rates of the transformants increased remarkably after 24 h of culture, all notably higher than that of Fv01. By observing the growth of the mycelium after recovery (Figure 4d), we noticed a markedly higher density of the transformants’ mycelia than that of the wild-type strain 48 h after recovery. Hence, *FfGS6* overexpression was presumed to be beneficial to mycelial repair.

### 3.5. The Action of FfGS6 in the Oxidative Stress Response of Mycelium

Oxidative stress was induced by adding H_2_O_2_ at different concentrations into the medium. After six days of culture, it was observed that the growth ability of the Fv01 mycelium was gradually weakened with the increase in H_2_O_2_ concentration. The treatment with 10 mmol/L H_2_O_2_ almost completely suppressed colonial growth (Figure 5a). The diameters of the colonies were further measured. As depicted in Figure 5b, the diameters of the Fv01 colonies decreased upon treatment with H_2_O_2_ in a concentration-dependent manner. Significant differences in diameter were noted when stimulated by 1.25 mmol/L H_2_O_2_. The RT-qPCR data also exhibited an inverse correlation of *FfGS6* gene expression with H_2_O_2_ concentration. The expression of *FfGS6* decreased as the H_2_O_2_ concentration increased without a significant difference (Figure 5c). The Pearson correlation analysis demonstrated a correlation coefficient of 0.93 between *FfGS6* expression and the diameters of the colonies, suggesting a strong positive correlation. It also showed a correlation coefficient of −0.79 between *FfGS6* expression and H_2_O_2_ concentration, indicating a strong negative correlation.

The significance of the *FfGS6* gene in the process of oxidative stress was clarified in the next step. Overexpression transformants were seeded onto the PDA medium replenished with 5 mmol/L H_2_O_2_, and the growth rate of the mycelium was measured. The growth rates of the transformants on the medium supplemented with H_2_O_2_ were considerably faster than that of the wild-type strain Fv01 (Figure 5d,e). With the growth rate of each strain on PDA medium used as the control, the inhibition rate of H_2_O_2_ on the mycelial growth was calculated (Figure 5f). Stimulation by 5 mmol/L H_2_O_2_ resulted in an inhibition rate of 62% on the Fv01 mycelial growth and inhibition rates lower than 50% on all four transformants. The results indicate that the *FfGS6* gene participated in the oxidative stress response process of *F. filiformis* mycelium and that its overexpression could prominently improve mycelium oxidative stress tolerance.

### 3.6. Impacts of FfGS6 Overexpression on F. filiformis Fruiting Body Development

Our previous works suggested that *FfGS6* was explicitly expressed on the stipe with the highest level in the elongation period [22]. To examine the role of the *FfGS6* gene in modulating fruit body development, two transformants (OE#10, and OE#16) with the most significant overexpression efficiencies were selected for fruiting experiments. Concurrently, the wild-type strain Fv01 served as the control. Morphological differences during fruit body development were recorded. As shown in Figure 6a, the mycelia of the material surfaces of the aforementioned two transformants completely turned white on the 42nd day of cultivation (five days after scratching the fungus), showing markedly stronger repairability than that of Fv01; on the 48th day of cultivation, all strains exhibited primordia formation, but the number of primordia in the two transformants was considerably less than that of Fv01; during fruit body development, the overexpression strains and Fv01 showed no noticeable difference in cap morphology but markedly different stipe elongation rates. Next, the stipe length was measured, and the results are summarized in Figure 6b. On the 57th day of cultivation, the stipe length of Fv01 reached 7.4 cm, while OE#10 and OE#16 had significantly shorter stipes (5.89 and 5.83 cm, respectively). Two overexpression transformants of *FfGS6* had the same decreasing trend of stipe. Therefore, the *FfGS6* gene meditated several characteristics, such as mycelial recovery ability, primordium number, and stipe elongation, during the fruiting body development of *F. filiformis*.

## 4. Discussion

Glucan is the most abundant fungal cell wall polysaccharide, representing 50–60% of the dry weight of this structure [30]. It contains two main β-glucans of β-1,3-glucan and β-1,6-glucan, and β-1,6-glucan has only been found in the cell wall of fungi and of members of the phylum Chromista, which may suggest a special role in fungi and Chromista biological processes [31]. Multiple studies are concerned with the function of β-1,3-glucan and its synthase. It has been demonstrated that β-1,3-glucan synthase is involved in cell wall integrity, the stress response, mycelium growth, and polysaccharide synthesis in filamentous fungi [32,33,34]. In *Metarhizium acridum*, FKS-RNAi transformants’ mycelia are more sensitive to the cell wall and hyperosmotic stress. In addition, aerial hyphae and conidial yield are obviously reduced in transformants on various media. The research on the β-1,3-glucan synthase gene (*GFGLS2*) of *Grifola frondosa* has shown that *GFGLS2* plays an important role in promoting the growth of mycelium and the synthesis of polysaccharides [32,34]. Several enzymes involved in β-1,6-glucan synthesis have recently been reported; however, the synthesis mechanism remains speculative [33].

In yeast, the genes encoding β-1,6-glucan synthases are essential for synthesizing β-1,6-glucan, and they act as essential potential functions in cell morphology and the stress response [17,35]. This also illustrates that the β-1,6-glucan synthesis-related protein KRE9 is vital for spore wall formation and that it regulates biological processes, such as yeast cell budding and mating [36,37]. Here, we obtained the β-1,6-glucan synthase gene (*FfGS6*) overexpressed transformants of *F. filiformis*, and we found that *FfGS6* overexpression could noticeably increase the cell width. This result affirms the β-1,6-glucan synthase gene function in cell morphology. Additionally, our study also showed that the *FfGS6* gene was involved in the response to both injury and oxidative stresses and that overexpressing *FfGS6* could significantly enhance the tolerance ability to abiotic stresses. These results provide evidence of the positive role of the *FfGS6* gene in the stress response.

A mushroom’s stipe is the primary edible part, and stipe elongation ability determines the product value of the mushroom. Numerous studies have shown that cell elongation triggered by rapid cell wall extension is the primary driver of stipe elongation in macrofungi [8,38]. Studies in *Coprinus cinereus* have illustrated that stipe elongation is induced after changes in the cell wall structure, and low concentrations of β-glucan-degrading enzymes can alter cell wall structure and induce stipe elongation [39,40]. In *A. bisporus*, β (1-6) glucan synthase gene expression was found to be up-regulated in the harvest fruiting bodies, and (1-6) β-linked glucan side branches also increased in the cell wall of rapid elongation, which indicated that the β (1-6) glucan synthase might be involved in cell wall composition changes [21]. Our previous study also found that *FfGS6* was differentially expressed in the different developmental stages of *F. filiformis* fruiting bodies. It markedly up-regulated expression on the stipe, exhibiting a significant positive correlation with the elongation speed of the stipe [22]. However, the results of this study show that the overexpression of *FfGS6* did not accelerate the stipe elongation of *F. filiformis* but significantly reduced the stipe elongation. This seems contrary to the relationship between *FfGS6* gene expression and the stipe elongation rate [22]. According to the results of Li et al. [41], the accumulation and cross-linking of excess β-1,6-glucan can lead to the loss of the stipe elongation ability of *C. cinereus*. We proposed that the expression of *FfGS6* may be necessary for cell wall synthesis during the stipe elongation of *F. filiformis*; however, a single elevation in cellular *FfGS6* expression may lead to the excessive accumulation of β-1,6-glucan, thereby inhibiting the stipe elongation of *F. filiformis*. Since stipe elongation is one of the most complex biological process of fungi and could be regulated by multiple genes [42,43]. Needless to say, more studies are needed to verify this speculation.

## 5. Conclusions

Our results suggest a potential role of *FfGS6* in regulating mycelial cell morphology, the mycelial stress response, and fruit body development in *F. filiformis*. These present studies will provide a foundation for a better understanding of the stress resistance and fruit body development of *F. filiformis* upon environmental stress.

## Figures and Tables

**Figure 1 genes-13-01753-f001:**
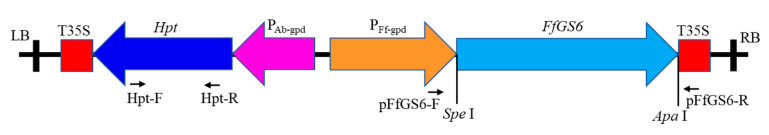
Schematic diagram of pBHg-*FfGS6*-OE recombinant plasmid T-DNA region. Hpt-F/Hpt-R, resistance gene amplification primer sites; pFfGS6-F/pFfGS6-R, target gene amplification primer sites; *Spe* I and *Apa* I, restriction sites.

**Figure 2 genes-13-01753-f002:**
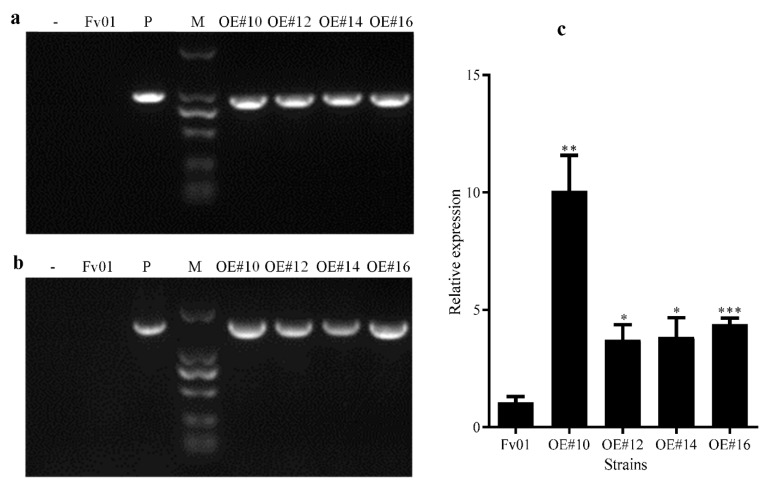
Verification of *FfGS6* overexpression transformants of *F. filiformis.* (**a**) PCR verification of *Hpt*-resistant gene sequences of overexpression transformants. M, Maker DL2000; -, blank control with water as a template; Fv01, wild-type strain; P, positive control plasmid; OE#10, OE#12, OE#14, and OE#16, *F. filiformis* overexpression transformants. (**b**) PCR verification of the *FfGS6* target gene sequence of the overexpression transformants. M, Maker DL2000; -, blank control with water as a template; Fv01, wild-type strain; P, positive control plasmid; OE#10, OE#12, OE#14, and OE#16, *F. filiformis* overexpression transformants. (**c**) qRT-PCR detection results of *FfGS6* gene expression levels in different overexpression transformants. Asterisks represent significant differences versus Fv01 (*t*-test, *n* = 3; *, *p* < 0.05; **, *p* < 0.01; ***, *p* < 0.001).

**Figure 3 genes-13-01753-f003:**
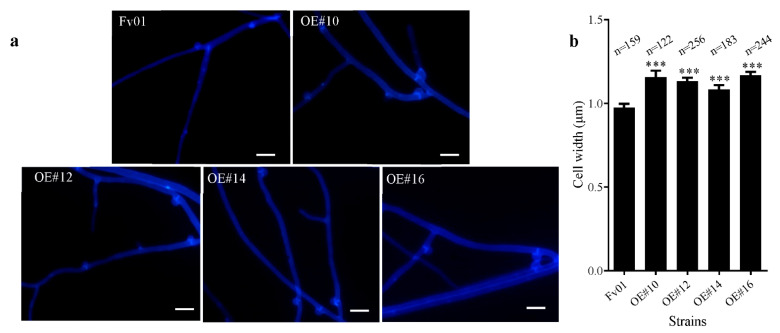
Microscopic observation of morphology and measurement of the cell width of *F. filiformis FfGS6* overexpression transformants. (**a**) Fluorescence microscopic observation of mycelium (scale bar = 5 μm). (**b**) Measurement of cell width at the clamp connection. Asterisks represent significant differences versus Fv01 (*t*-test, *n* ≥ 110; ***, *p* < 0.001).

**Figure 4 genes-13-01753-f004:**
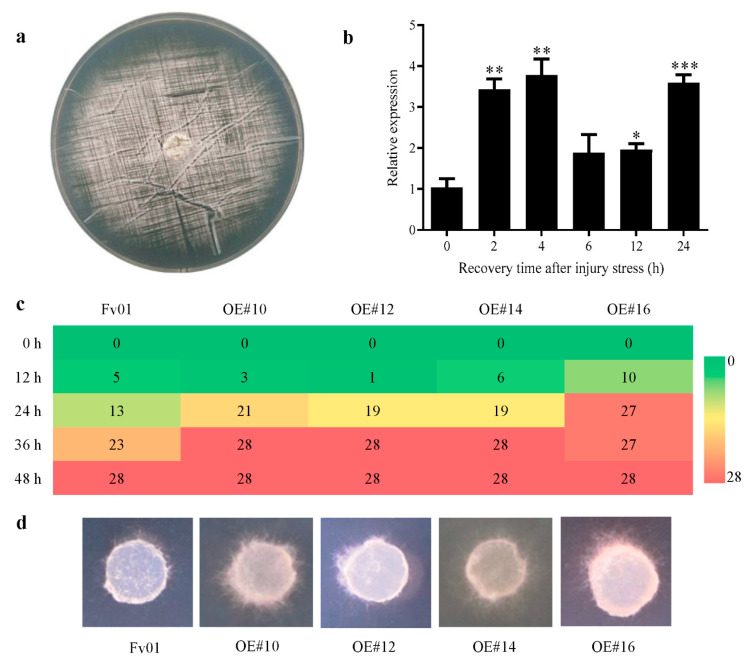
The expression pattern of *F. filiformis FfGS6* in response to injury stress and mycelial repairability of its overexpression transformants. (**a**) *F. filiformis* Fv01 mycelial injury stress induction; (**b**) *FfGS6* gene expression at different recovery time points (0, 2, 4, 6, 12, and 24 h) after Fv01 mycelial injury induction. Asterisks represent significant differences versus Fv01 (*t*-test, *n* = 3; *, *p* < 0.05; **, *p* < 0.01; ***, *p* < 0.001). (**c**) Recovery ability assessment within 48 h after mycelial injury. (**d**) The repair of mycelium after 48 h of injury stress induction.

**Figure 5 genes-13-01753-f005:**
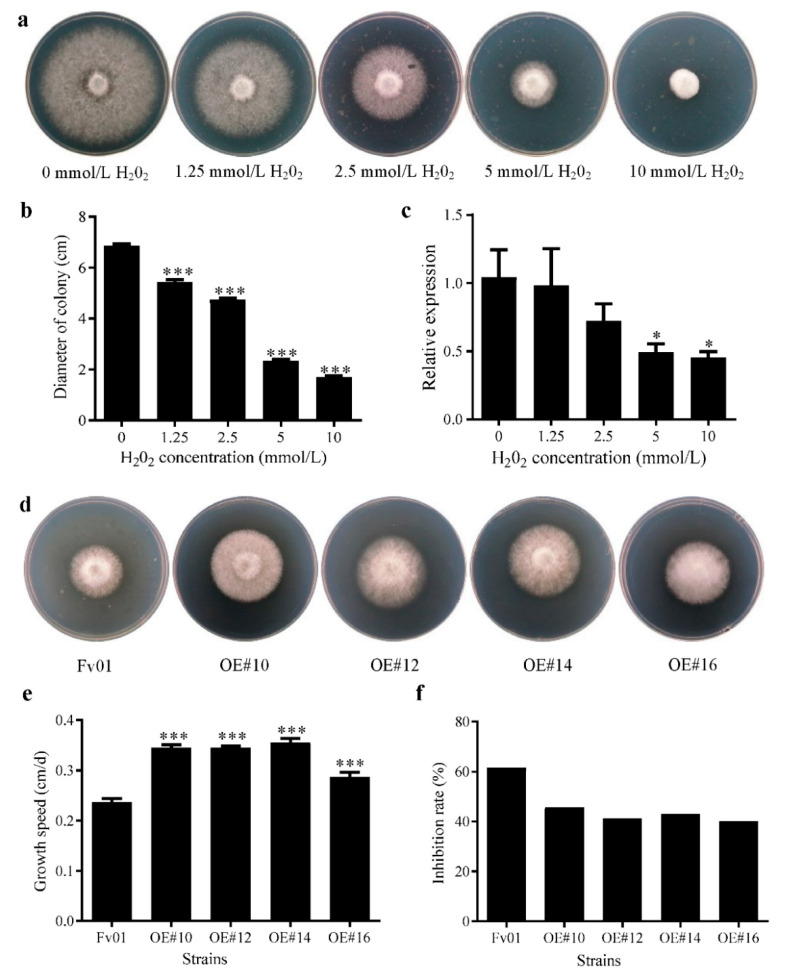
The expression of *F. filiformis FfGS6* in response to H_2_O_2_ stress and the growth inhibition rate of H_2_O_2_ on the overexpression transformants. (**a**) The growth phenotype of wild-type strain Fv01 was cultured in a medium containing different concentrations (0, 1.25, 2.5, 5, and 10 mmol/L) of H_2_O_2_. (**b**) The diameters of colonies after wild-type strain Fv01 was cultured in the medium containing different concentrations (0, 1.25, 2.5, 5, and 10 mmol/L) of H_2_O_2_. (**c**) *FfGS6* gene expression after wild-type strain Fv01 was cultured in the medium containing different concentrations (0, 1.25, 2.5, 5, and 10 mmol/L) of H_2_O_2_. (**d**) Phenotypic observation of Fv01 and *FfGS6* overexpression transformants cultured on 5 mmol/L H_2_O_2_ PDA plate. (**e**) Mycelial growth rates of Fv01 and *FfGS6* overexpression transformants cultured in 5 mmol/L H_2_O_2_ PDA plates. (**f**) Growth inhibition rates of Fv01 and *FfGS6* overexpression transformants on 5 mmol/L H_2_O_2_ PDA plates. Asterisks represent the significant differences versus 0 mol/L H_2_O_2_ (*t*-test, *n* = 3; *, *p* < 0.05; ***, *p* < 0.001) and Fv01.

**Figure 6 genes-13-01753-f006:**
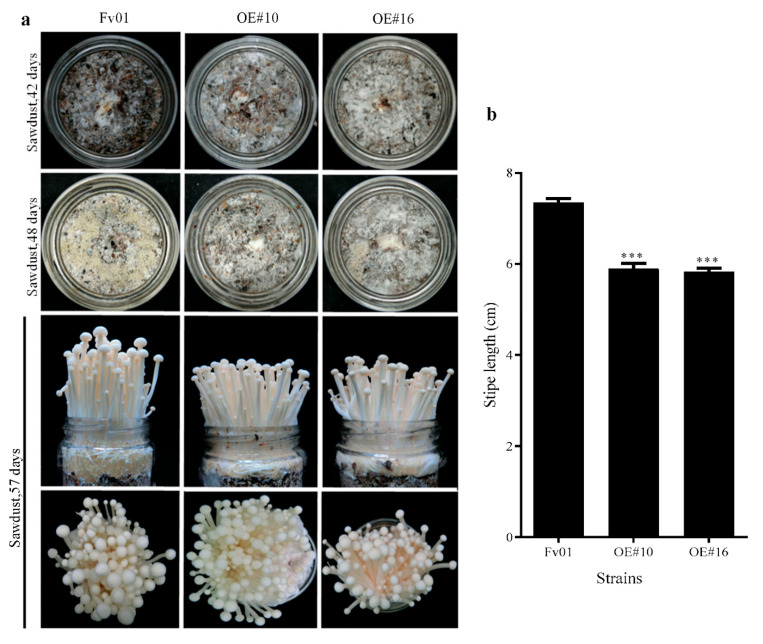
The fruiting body phenotype of transformants. (**a**) Fruiting phenotype of *FfGS6* overexpression transformants of *F. filiformis* and the control group. (**b**) Stipe length measurement 9 days after primordium formation. Asterisks represent significant differences versus Fv01 (*t*-test, *n* = 20; ***, *p* < 0.001).

**Table 1 genes-13-01753-t001:** Primer sequences.

Primer Name	Primer Sequence (5′-3′)	Purpose
pFfGS6-F	CATCTGCTGTTTGCTGCTC	Verify target fragment
pFfGS6-R	CCCTTATCTGGGAACTACTCAC	
Hpt-F	CACATCCACCATCTCCGT	Amplify resistance gene
Hpt-R	AAATTGCCGTCAACCAAG	
qFfGAPDH-F	CCTCTGCTCACTTGAAGGGT	qRT-PCR
qFfGAPDH-R	GCGTTGGAGATGACTTTGAA	
qFfGS6-F	GAGTTGTGGTCGTTAAAGGGAA	qRT-PCR
qFfGS6-R	CGTCAATCATATCCACAGCGT	

## Data Availability

The data presented in this study are available in the manuscript and Appendix A.

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
