# Peer review of "Effects of β-1,6-Glucan Synthase Gene (FfGS6) Overexpression on Stress Response and Fruit Body Development in Flammulina filiformis"

_genes, 2022, doi:10.3390/genes13101753_

Round 1
Reviewer 1 Report
MDPI Genes MS ID: 1917983
Effects of β-1,6-Glucan Synthase Gene (FfGS6) Overexpression 2 on Stress Response and Fruit Body Development in Flammul- 3 ina filiformis
General comments
In the present study the authors have shown that the overexpression transformants of β-1, 6-glucan synthase gene (FfGS6) in Flammulina filiformis improve the mycelium width and its ability to tolerate mechanical injury and oxidative stress. The MS is well written, however there are some concerns
i. The rephrasing of the sentences are required at many places throughout the MS to improve the readability.
ii. There is lack of proper controls in the mechanical injury experiment. In this experimat’s results the expression pattern shows that the expression of the gene is occurring in cycles. This type of expression is also typical of normal growth when genes are expressed during elongation/expansion of the cell wall. Therefore, including 'uninjured' respective controls for each strain will give real picture, if the expression pattern is normal or responsive to external mechanical injury.
The authors are advised to include these controls in your results.
Reviewer 2 Report
This study leads us closer to the proper understanding of biological role and action mechanism of Glucan, being the most abundant fungal cell wall polysaccharide. My question are as follows:
- In the paper to be published four overexpression transformants of β-1, 6-glucan synthase gene were obtained. Is it destined and planned to study gene function in different species apart from Flammulina filiformis?
- How could it be more comprehensively illustrated that β-1,3-glucan synthase is involved in cell wall integrity, stresses response, and mycelium growth in filamentous fungi? Could you involve more references in this respect?
- How may β-1,6-glucan synthesis be stimulated by means of other treatments/conditions? Could you involve some relevant such procedures?
- What kind of other roles might be attributed to FfGS6 gene in the stress response?
Reviewer 3 Report
-
Enoki mushroom is the second largest mushroom industry in China. The shape of the stipe and pileus of this mushroom affects market sales. Therefore, it is very meaningful for the author to study the growth and development of F. filiformis through β-glucan synthase. Although the findings resulted in a shortened stipe, this does not detract from the importance of this manuscript. However, I must concern something, in the overexpression strain, the stipe is shorter but thicker. Will the proportion of hollow inside these stipes become lower? The hollowness of the stipe of this mushroom also affects the taste.
- Minors:
-
L95: Italics are not required for primers.
-
Please check throughout the text
- L125: ℃ should be revised as °C
- L144:ΔΔCt should be in italics
- L207: Transformant revised as Transformants
- L249: t of t-test, and n should be in italics.
-
